# In Vitro Wound Healing Potential of a Fibroin Film Incorporating a Cannabidiol/2-Hydroxypropyl-β-cyclodextrin Complex

**DOI:** 10.3390/pharmaceutics15122682

**Published:** 2023-11-27

**Authors:** Thamonphat Klinsang, Pensri Charoensit, Preeyawass Phimnuan, Kunlathida Luangpraditkun, Gareth M. Ross, Céline Viennet, Sukunya Ross, Jarupa Viyoch

**Affiliations:** 1Department of Pharmaceutical Technology, Faculty of Pharmaceutical Sciences, Center of Excellence for Innovation in Chemistry, Naresuan University, Phitsanulok 65000, Thailand; thamonphatk62@nu.ac.th (T.K.); pensric@nu.ac.th (P.C.); 2Health Intervention and Technology Assessment Program, Department of Health, Ministry of Public Health, Nonthaburi 11000, Thailand; preeyawass.p@hitap.net; 3Research Unit of Pharmaceutical Innovations of Natural Products (PhInNat), Burapha University, Chonburi 20131, Thailand; kunlathida.lu@go.buu.ac.th; 4Department of Chemistry, Center of Excellence in Biomaterials, Faculty of Science, Naresuan University, Phitsanulok 65000, Thailand; gareth@nu.ac.th; 5UMR 1098 RIGHT INSERM EFS FC, DImaCell Imaging Resource Center, University of Franche-Comté, 25000 Besançon, France; celine.viennet@univ-fcomte.fr

**Keywords:** fibroin, cannabidiol, CBD/hydroxypropyl-β-cyclodextrin complex, biomaterial, wound healing, fibroblasts

## Abstract

This study aimed to develop a film dressing prepared by incorporating a complex of cannabidiol and 2-hydroxypropyl-β-cyclodextrin (CBD/HP-β-CD) into a fibroin-based film and to investigate its wound healing capabilities. The fibroin from silkworm cocoons exhibited a total protein content of 96.34 ± 0.14% *w*/*w* and a molecular weight range of 25 to 245 kDa. Fourier-transform infrared spectroscopy (FTIR) revealed the presence of characteristic amide peaks (I, II, and III) in the isolated fibroin. The CBD/HP-β-CD complex, prepared with a molar ratio of 1:2 (CBD to HP-β-CD), had 81.5 ± 1.2% *w*/*w* CBD content, as determined by high-performance liquid chromatography (HPLC). X-ray diffraction (XRD) and FTIR analyses demonstrated successful encapsulation of CBD’s hydrophobic aromatic rings by HP-β-CD. Blending the fibroin solution with the CBD/HP-β-CD complex produced a transparent, slightly yellowish film. Mechanical testing revealed a tensile strength of 48.67 ± 2.57 MPa and a % elongation at a break of 1.71 ± 0.21%. XRD and FTIR analyses showed distinctive crystalline and chemical structures of the film. In subsequent in vitro experiments with normal human dermal fibroblasts, the film demonstrated potential for wound healing. An increase in cell division (G2/M phase) was observed compared to the fibroin film without the CBD/HP-β-CD complex. Additionally, fibroblasts treated with the film exhibited enhanced cell migration in a scratch assay and increased expression of vascular endothelial growth factor protein compared to the control group. Overall, these findings underscore the film’s potential for enhancing wound healing outcomes.

## 1. Introduction

Wound healing is a complex and dynamic process that involves four principal phases: (1) hemostasis, which includes blood vessel constriction and fibrin clot formation; (2) inflammation, with inflammatory cells migrating to the wound site; (3) migration and proliferation of cells related to granulation tissue formation and angiogenesis; and (4) remodeling of the extracellular matrix (ECM) to enhance the integrity and functionality of the new tissue. Throughout this process, various types of cells, mediators, and cytokines collaborate harmoniously. These cells include keratinocytes, fibroblasts, endothelial cells, neutrophils, macrophages, and lymphocytes, all of which operate and coordinate in a precise sequence and duration to complete wound healing. The operation and coordination of these cells rely on autocrine and paracrine signals based on the release of pro-inflammatory cytokines and growth factors, such as basic fibroblast growth factors (bFGF), transforming growth factor β (TGF-β), and vascular endothelial growth factors (VEGF) [1,2].

Fibroblasts play a pivotal role in all phases of wound healing. Upon migrating into the wound, they undergo activation and differentiate into myofibroblasts. This differentiation provokes them to generate the extracellular matrix (ECM). Another function of fibroblasts is the contribution to the production and secretion of VEGF [2,3], an essential mediator in cell migration, angiogenesis, granulation tissue formation, and epidermal closure. The formation of granulation tissue usually occurs in close association with the inflammatory phase, and the two events may overlap. However, it is crucial for the inflammatory phase to be appropriately regulated and not excessively prolonged. Prolonged inflammation is associated with increased production and activity of matrix metalloproteases [4,5], which are proteolytic enzymes capable of degrading collagen and other extracellular components of the granulation tissue. Hence, long-term inflammation can lead to the persistence of chronic non-healing wounds [4].

Film and hydrogel dressings are commonly used devices for wound protection. To enable effective wound healing and minimize possible cutaneous complications, the dressing should possess specific biological activities, such as free radical scavenging, anti-inflammatory, pro-angiogenic, and induction of cell growth and migration activities. These biological activities can be attributed to compounds found in the dressing, whether they are synthetic, semi-synthetic, or naturally occurring. A report on successful wound healing in diabetic rats using a hydrogel incorporated with a compound synthesized from melanin/gold/platinum that facilitates hyperthermia and disrupts the ROS-inflammation pathway [6] is one example. Nevertheless, persisting challenges encompass complex synthesis, cost implications, and environmental considerations during the production process.

Silk fibroin is a naturally derived biomaterial that has demonstrated promising results in wound healing. Our previous studies have demonstrated that fibroin isolated from silk cocoons using a simple and eco-friendly technique can effectively induce cell proliferation and migration [7,8]. These stimulatory effects are likely mediated through the activation of mitogen-activated protein kinases/extracellular signal-regulated kinase (MAPK/ERK) pathways [7]. Moreover, silk fibroin exhibits angiogenic activity, contributing to quicker skin wound healing [9,10]. Silk fibroin-based films also serve as a suitable matrix for the incorporation of additional components, synergizing the healing process [11,12]. One such component that has attracted our attention for wound dressing applications is cannabidiol (CBD). CBD has drawn interest due to its multifunctional activities, including anti-inflammatory, pain relief, and antioxidant properties, without inducing psychoactive effects. Enhancing wound healing properties of CBD has been demonstrated in both in vitro and in vivo studies [13]. The mechanism of action of CBD in wound healing is believed to be associated with the endocannabinoid system and the cannabidiol receptors type-2 expressed on skin cells [14].

According to the interesting therapeutic potential of both fibroin and CBD, our objective was to develop a dressing film by incorporating CBD into a fibroin film to synergistically enhance wound healing activity. However, a significant limitation of its application was the low solubility and bioavailability of CBD. To overcome this challenge, we employed a complexation technique entrapping CBD into 2-hydroxypropyl-β-cyclodextrin (HP-β-CD) to enhance the solubility of CBD in an aqueous solution of fibroin. Compared to β-CD, HP-β-CD offers enhanced solubility in water (up to 600 mg/mL [15]) and stability, along with a decrease in toxicity [16]. After casting, the dried fibroin film served as a polymeric matrix for the dispersed CBD/HP-β-CD complex. We evaluated the physicochemical properties of the developed film to ensure its suitability for handling and to assess the interaction of the CBD/HP-β-CD complex with the fibroin matrix. Furthermore, we examined the biological activities of the developed film on normal human dermal fibroblast (NHDF) cells, including cell toxicity, proliferation, migration, and VEGF expression. The findings from our study demonstrate the potential of the developed film in enhancing wound healing. By combining the benefits of CBD and fibroin and employing the complexation technique to improve CBD solubility, we have taken a significant step towards the development of an effective wound dressing film.

## 2. Results

### 2.1. Characteristics of the Isolated Fibroin

The isolated fibroin from silkworm cocoons (Nang-Lai strain) exhibited a yellowish and cotton-like appearance, as shown in Figure 1A. The isolation process resulted in an approximate yield of 68% *w*/*w* from the cocoons, and the isolated fibroin was found to contain a total protein content of 96.34 ± 0.14% *w*/*w*. The SDS-PAGE technique revealed two distinct bands of proteins (Figure 1B): a smear band with molecular weights ranging from 30 to 245 kDa and a specific band at approximately 25 kDa. The FTIR results are displayed in Figure 1C. The characteristic peaks observed for the amide groups were as follows: amide I exhibited a frequency peak ranging from 1696 to 1611 cm^−1^, amide II showed peaks between 1550 and 1501 cm^−1^, and amide III displayed peaks in the range of 1400 to 1200 cm^−1^. Furthermore, N-H stretching, -OH stretching, and strong hydrogen bonding were observed in the region of 3600 to 3200 cm^−1^.

### 2.2. Characteristics of the CBD/HP-β-CD Inclusion Complex

In Figure 2A, the appearance of the CBD/HP-β-CD complex is depicted as a transparent and white crystalline solid. The CBD content within the complex was found to be 81.5 ± 1.2% *w/w*, as determined by the HPLC calibration curve generated from standard CBD. The FE-SEM technique was utilized to investigate the morphology before and after inclusion. The FE-SEM images of CBD, HP-β-CD, and the CBD/HP-β-CD complex are presented in Figure 2B. CBD appeared with a crystalline structure characterized by nubby shapes of varying diameters, while HP-β-CD exhibited a spherical structure of various sizes. In contrast, the complex displayed an irregular blocky structure, distinguishing it from the structures of CBD and HP-β-CD. Focusing on their crystalline structures as determined by XRD, the individual XRD patterns of CBD, HP-β-CD, and the inclusion complex are presented in Figure 2C. In the XRD pattern of CBD, distinct peaks were observed at diffraction degrees of 14.22, 15.37, 20.58, 22.13, 25.20, and 34.84, while HP-β-CD showed a broad diffraction peak ranging from 15.00 to 21.00 degrees. Notably, the XRD pattern of the CBD/HP-β-CD inclusion complex differed significantly from those of CBD and HP-β-CD. The inclusion complex did not simply represent a superposition of CBD and HP-β-CD, in which some crystalline peaks of CBD, such as those at 14.22, 22.13, 25.20, and 34.84 degrees, disappeared. Moreover, the formation of two new diffraction peaks at 16.20 and 18.60 degrees was also observed.

The IR spectra results of CBD, HP-β-CD, and the CBD/HP-β-CD inclusion complex are demonstrated in Figure 2D. The FTIR characteristic peaks of CBD were mainly observed at 3550 to 3200 cm^−1^, representing -OH stretching and strong hydrogen bond, and at 3080 cm^−1^, indicating -C-H stretching of aromatic rings. Additional peaks at 2960, 2927, 2850, and 2825 cm^−1^ corresponded to the -C-H stretching of methyl or methylene groups. CBD’s characteristic -C=C stretching in the aromatic ring was observed at 1646, 1605, and 1531 cm^−1^. Additional peaks at 1460 and 1240 cm^−1^ were also observed, corresponding to -OH bending and -C-O stretching, respectively. For HP-β-CD, a broad peak of -OH stretching and strong hydrogen bonds were observed at 3650 to 3000 cm^−1^. The -C-H stretching of methyl or methylene groups appeared at the same wavenumber as CBD but was broader. Peaks of C-H bending of alkane, C-O stretching, and a strong peak of C-O-C stretching were observed at 1500–1238, 1175, and 1042 cm^−1^, respectively. In the CBD/HP-β-CD complex, broad peaks of -OH stretching and strong hydrogen bonding were observed at 3650 to 3000 cm^−1^, primarily from HP-β-CD. A peak of C-H stretching was observed in the combined frequency ranges of CBD and HP-β-CD. The peaks of C=O stretching (CBD), -OH bending (CBD), -C-H bending of alkane (HP-β-CD), and the peak of C-O-C stretching (HP-β-CD) were observed at the same frequency ranges as either CBD or HP-β-CD. However, the peak of -C-H stretching of the aromatic rings of CBD at 3080 cm^−1^ disappeared from the complex.

### 2.3. Characteristics of the Fibroin Film Containing the CBD/HP-β-CD Complex

An appearance of the developed film (fibroin film containing the CBD/HP-β-CD complex) is presented in Figure 3A. The film, fabricated through the casting technique, exhibited a transparent and slightly yellowish appearance, with a uniform thickness of 50 ± 5 µm. The average amount of CBD in the film was 100 ± 11 µg/cm^2^. Figure 3B shows the surface and cross-section of the developed film, which exhibits a compact (non-porous) and relatively homogenous structure. Regarding the mechanical properties, the developed film demonstrated a tensile strength at a maximum force of 48.67 ± 2.57 MPa and 1.71 ± 0.21% for the % elongation at break, while the control film (fibroin film without the complex) exhibited 46.14 ± 2.81 MPa for the maximum force and 1.51 ± 0.26 for the % elongation at break, as shown in Figure 3C. Figure 3D presents the XRD pattern of the isolated fibroin, the CBD/HP-β-CD complex, and the developed film. No crystalline peaks were observed in the isolated fibroin. However, in the developed film, a new crystalline characteristic peak was observed at approximately 21 degrees, while the crystalline peaks of the CBD/HP-β-CD complex at 16.20 and 18.60 degrees disappeared. Figure 3E demonstrates the FTIR spectra of the developed film in comparison to those of the isolated fibroin and the CBD/HP-β-CD complex, covering the frequency range from 4000 to 400 cm^−1^. The developed film displayed characteristics of both isolated fibroin and the CBD/HP-β-CD complex. The N-H stretching, -OH stretching, and strong hydrogen bond peak were observed as a combination of the broad frequency peaks of isolated fibroin and CBD at 3650 to 3000 cm^−1^. Interestingly, broader peaks of amide I and amide II, along with shifts to higher wavenumbers of amide III and C-O-C stretching, were observed in the developed film.

### 2.4. Wound Healing Potential of the Developed Film (Fibroin Film Containing the CBD/HP-β-CD Complex)

The cytocompatibility of the developed film was initially examined on NHDF cells. The obtained results on cell viability and morphology strongly suggest that the developed film had no negative effects on cell viability but, in contrast, increased the proliferation of the NHDF cells, as shown in Figure 4A,B. By comparing the proliferation rates of cells treated with the control film (the fibroin film without the complex) and the developed film-free supernatant to those of control (cells treated with serum/antibiotic-free DMEM), the control film and developed film significantly enhanced proliferation (108.4 ± 2.1%, *p* < 0.05 and 146.5 ± 5.3, *p* < 0.01.

Figure 5 illustrates the cell cycle results. Cells treated with the developed film-free supernatant exhibited a significant (*p* < 0.01) increase in the phase of complete DNA synthesis and cell division (G2/M phase), reaching 25.0 ± 2.1%, compared to that of the control film (20.1 ± 1.0%). These values were markedly higher compared to the control group, which showed 10.9 ± 0.7%.

The immunofluorescence assay was conducted to assess the expression of VEGF protein in NHDF. We observed a higher expression of VEGF protein in fibroblasts treated with either the fibroin film or the developed film-free supernatant in comparison to the control group, as shown in Figure 6.

Finally, the beneficial effect of the developed film on wound management was assessed using a wound healing test or scratch assay in a 2-well insert. The recolonization of the gap area was monitored under an optical microscope. We observed that the gap in the NHDF cells treated with the developed film-free supernatant was almost completely healed at 36 h after gap zone creation, whereas the control group did not show healing simultaneously, as shown in Figure 7.

## 3. Discussion

Both fibroin and CBD exhibit noteworthy biological activities that promote wound healing. Therefore, we expected that a film cast using these components should also possess beneficial wound-healing properties. To overcome the limitation of CBD’s solubility in the aqueous casting solution of fibroin, a complex between CBD and HP-β-CD was initially prepared. We successfully incorporate the CBD/HP-β-CD complex into the fibroin matrix, resulting in distinct crystalline and chemical structures of the cast film and enhanced wound healing via in vitro scratch assay using normal human dermal fibroblast (NHDF) cells.

Our work started with the isolation of fibroin from silk cocoons. We observed that the percentage yield of the isolated fibroin and the protein content (approximately 96% *w/w*) in the isolated fibroin were consistent with our previous studies [7,8,17]. Using the SDS-PAGE technique, we observed that the fibroin protein consisted of a broad smear and specific bands. The presence of the broad smear bands may indicate the degradation of the heavy chain (325–350 kDa) of silk fibroin [18], while the specific band at 25 kDa is likely attributed to a light chain of fibroin [7,8,19]. FTIR spectroscopy was utilized to determine the chemical functional groups present in the isolated fibroin, covering a wavelength range of 4000 to 400 cm^−1^. As silk fibroin is a protein consisting of a sequence of amino acids, it contains distinct amide groups on the backbone, namely amide I (C=O stretching), amide II (N-H bending), and amide III (in-phase combination of N-H in-plane bending and C-N stretching) [7,8,18]. In addition, a distinct board peak, corresponding to N-H stretching, -OH stretching, and strong hydrogen bonding within the protein chains of the isolated fibroin, was found.

In order to improve the aqueous solubility of CBD, it was prepared in an inclusion complex with HP-β-CD. The results from HPLC analysis showed a high content of entrapped CBD (about 82% *w/w*), which indicates the effectiveness of our inclusion technique in entrapping CBD. Results from FE-SEM revealed a complete transformation in the appearance of both CBD and HP-β-CD structures within the CBD/HP-β-CD complex, resulting in a homogeneous and irregular blocky structure. The successful formation of the complex was also evident in different XRD patterns and IR spectra [20,21] compared to the individual amorphous HP-β-CD and crystalline CBD. As complexation disrupted the crystalline nature of poorly water-soluble molecules [22], several crystalline peaks of CBD disappeared (at 14.22, 22.13, 25.20, and 34.84 degrees), while two new diffraction peaks were formed (at 16.20 and 18.60 degrees). Furthermore, the results from the IR spectra confirmed the successful formation of the CBD/HP-β-CD complex by showing shifts or disappearances of peaks from each parent compound [23]. Notably, C=O stretching (CBD), -OH bending (CBD), -C-H bending of alkane (HP-β-CD) peaks, and the peak of C-O-C stretching (HP-β-CD) were observed at the same frequency ranges as in either parent CBD or HP-β-CD. However, the peak of -C-H stretching of the aromatic rings of CBD at 3080 cm^−1^ disappeared in the complex. These findings suggest that CBD, particularly the hydrophobic characteristic of its aromatic rings, was entirely entrapped by HP-β-CD [24], thereby restricting the vibration of functional groups within CBD molecules. Once again, the observed changes in the appearance, XRD pattern, and FTIR spectrum provide strong evidence of significant structural alterations and molecular interactions between CBD and HP-β-CD, supporting the formation of the inclusion complex and its potential application in enhancing CBD’s solubility [21,24].

As previously mentioned, our results demonstrate the effective entrapment of the hydrophobic CBD moiety within the lipophilicity cavity of HP-β-CD; this allows the hydrophilic outer surface of HP-β-CD to remain compatible with the aqueous solution of fibroin. The uniformity exhibited by the film indicates excellent compatibility between the CBD/HP-β-CD complex and the fibroin matrix solution. Furthermore, the presence of CBD confirms the accomplished incorporation of the complex into the film matrix. The XRD pattern of the developed film revealed a distinctive crystalline peak at approximately 21 degrees, which was shifted from the peak of the neat complex at 16.20 and 18.60 degrees. In the FTIR spectrum, the broader and shifted peaks observed in the developed film may be attributed to the formation of hydrogen bonding between the isolated fibroin and the complex, resulting in changes in the electron cloud and the resonant frequencies of the bonds [25,26]. Consistently, all the XRD and FTIR findings confirm the successful incorporation and interaction of the CBD/HP-β-CD complex with the fibroin matrix, leading to notable modifications in both crystalline and chemical structures. This synergistic combination of components within the film matrix offers unique properties. When examining the cross-sectional view of the developed film using FE-SEM, we observed ductile and nanometer-scaled domains. The irregularity in these ductile domains indicates the fibrillar nature of the internal assembly within the film. The non-porous structure of the film resulted in relatively low flexibility [27]. However, the incorporation of lactic acid into the casting solution had a plasticizing effect [7,8,17], which improved the film’s flexibility to a level suitable for handling. It is important to note that pure fibroin film is typically brittle and unsuitable for practical applications, but in this study, our modification rendered it more flexible and useful.

In this study, we did not investigate the anti-inflammatory activity of the developed film using inflammatory cells. Instead, our primary focus was on investigating the potential use of the developed film in dermal fibroblast cultures, as fibroblasts play a central role in the wound healing process [2]. We found that the incorporation of the CBD/HP-β-CD complex into the fibroin film markedly enhanced the proliferation of the cells. These findings align with a previous study indicating CBD’s dose-dependent promotion of human dermal fibroblast growth [28] and prove our hypothesis on the synergistic interplay between fibroin and CBD. Regarding cell proliferation in relation to the cell cycle, these results are in concordance with our previous study in which fibroin exerted potential effects on cell division or G2/M phase [8], and they also align with the proliferation results obtained in this current study. Generally, inflammation can disrupt the normal cell cycle and promote abnormal cell growth [29]. The anti-inflammatory activity of CBD [30,31] may help restore normal cell cycle regulation, promoting healthy cell division and preventing uncontrolled cell proliferation.

Fibroblasts play a crucial role in the healing cascade events by secreting various growth factors, including basic fibroblast growth factor (bFGF), transforming growth factor β (TGF-β), and VEGF. Among these factors, VEGF specifically plays a key role in the migration process of fibroblasts and endothelial cells [2,3,32]. Notably, a recent study revealed that overexpression of VEGF in dermal fibroblasts accelerated wound healing functions, including the formation of granulation tissue, collagen deposition, and angiogenesis [2]. Consequently, in our present study, we investigated the expression of VEGF in fibroblasts treated with fibroin and developed film. It can be inferred that the augmented cell proliferation induced by the film is, at least in part, attributed to the activation of VEGF production and subsequent release, leading to an autocrine impact on fibroblasts. The precise mechanism underlying the induction of VEGF expression by the fibroin film containing the CBD/HP-β-CD complex requires further investigation. It is plausible that this mechanism may involve the activation of the MAPK/ERK signaling pathway by fibroin [7] and/or pathways related to the anti-inflammatory activity of CBD [30,31].

Finally, a scratch assay was conducted, as this method mimics a wound and assesses two-dimensional *cell* migration activity. The obtained results demonstrated that the developed film promoted an increase in cell growth and migration, which may be linked to the enhanced expression of VEFG, as described earlier. It is important to note, however, that a prior study reported that CBD inhibited the migration of human umbilical vein endothelial cells (HUVEC) in a concentration-dependent manner [33]. The observed discrepancy in results could be attributed to the variation in the distribution of cannabinoid receptors and the signaling responses between fibroblasts and HUVEC. Additionally, in a rat model, a hydrogel containing CBD was found to promote higher epidermis regeneration, collagen deposition, and the development of granulation tissue while reducing inflammatory cell levels [13]. All findings collectively suggest the potential of the fibroin film containing the CBD/HP-β-CD complex to contribute positively to the wound healing process by promoting cell proliferation, migration, and VEGF expression. Furthermore, the results suggest that the entrapped CBD within the complex and film matrix could dissolve and subsequently be released, thereby exhibiting its biological activities on the cells.

Delayed wound healing is a significant clinical challenge that carries substantial healthcare costs. Therefore, wound dressings that are both cost-effective and environmentally friendly and that facilitate wound healing would offer valuable benefits. To confirm the efficacy and benefits of the developed film, a clinical study comparing it with an available natural polymer dressing, such as collagen wound dressings, should be conducted in the future to assess their healing properties and potential adverse effects.

## 4. Materials and Methods

### 4.1. Materials

Silkworm cocoons (*Bombyx mori* Linn., Nang-Lai strain) were kindly supplied by the Queen Sirikit Sericulture Center, Chiang Mai. Cannabidiol (CBD, purity ≥ 95%, MW 314.5), a crystalline solid, was supplied by Cayman Chemical, MI, USA. 2-hydroxypropyl-β-cyclodextrin (HP-β-CD) was purchased from Sigma-Aldrich Co., Rockville, MD, USA.

### 4.2. Preparation of Silk Fibroin

The process for the preparation of silk fibroin was modified based on the previously reported protocols [7,8,17,19]. Briefly, the impurities were removed from the cocoon, which was then cut into small pieces. The pieces of the silk cocoon were boiled at 85–90 °C for 2 h. The boiled silk cocoon was then degummed by boiling it in 25 mM NaOH (RCI Labscan, Bangkok, Thailand) for 30 min at 70 °C. The degummed silk cocoon was rinsed three times with distilled water before drying in an oven at 50 °C overnight and then boiled in 3 M CaCl_2_ (RCI Labscan) at 85–90 °C for 4–6 h at a ratio of 0.25 g degummed silk cocoon per 15 mL CaCl_2_ solution. The resultant solution was dialyzed in 15 MΩ ultrapure water using molecular porous membrane tubing (MWCO: 6000–8000 Dalton) for 48 h (water was changed every 4–6 h). The dialyzed solution was then centrifuged at 8000 rpm at 4 °C for 15 min to remove insoluble debris. The clear solution was lyophilized by a freeze-drying procedure for 72 h to produce the isolated fibroin, which was kept dry until use.

### 4.3. Characterization of the Isolated Fibroin

#### 4.3.1. Determination of the Protein Content

The protein content of the isolated fibroin was determined using a DC protein assay kit (BIO-RAD Laboratories, Philadelphia, PA, USA), following the manufacturer’s instructions. Bovine serum albumin (BSA, Sigma-Aldrich Co., Burlington, MA, USA) was used as the reference protein. The reference protein and sample were incubated in the dark at room temperature (RT) for 15 min and then measured at an absorbance of 750 nm by microplate spectrophotometer (Eon^TM^, BioTek Instrument, Winooski, VT, USA).

#### 4.3.2. Determination of the Molecular Weight Pattern

Protein in the isolated fibroin was separated using the sodium dodecyl sulfate-polyacrylamide gel electrophoresis (SDS-PAGE) method on a 5% stacking gel and a 12% separating gel. After electrophoresis, the gel was stained in Coomassie Brilliant Blue R-250 solution (BIO-RAD Laboratories) for 2 h and de-stained with a mixture of 7% acetic acid (VWR International Ltd., Poole, UK) and 10% methanol (RCI Labscan) in water. The molecular weights of protein bands were estimated by comparing them with standard molecular weight markers.

#### 4.3.3. Chemical Functional Groups

The functional groups of the isolated fibroin were observed by Fourier transform infrared spectroscopy (FTIR, Perkin Elmer Spectrum GX series, Waltham, MA, USA). The spectra were scanned over wavenumbers ranging from 4000 to 400 cm^−1^.

### 4.4. Preparation of the CBD/HP-β-CD Inclusion Complex

CBD and HP-β-CD were weighed in a 1:2 molar ratio and then solubilized in 40% ethanol (Merck KGaA, Darmstadt, Germany). The CBD solution was slowly dropwise into the HP-β-CD solution while continuously stirring for 1 h. Afterward, the mixture was subjected to lyophilization using freeze-drying for 48 h, and the resulting product (CBD/HP-β-CD complex) was kept dry until it was ready for use.

### 4.5. Characterization of the CBD/HP-β-CD Inclusion Complex

#### 4.5.1. The CBD Content in the Complex

The quantity of CBD in the complex was determined through high-performance liquid chromatography (HPLC) using an Agilent 1260 Infinity II LC System (Palo Alto, CA, USA) with G7129A 1260 vial sampler, G711A 1260 Quat pump VL, and G7115A 1260 DAD WR, along with the LC Open Lab software (offline). The detection wavelength was set at 220 nm, and a C18-packed column of 2.6 μm particle size and dimensions of 4.6 × 150 mm was used. For the mobile phase, a mixture of acetonitrile (RCI Labscan) and 20 mM ammonium formate (Sigma-Aldrich Co., Waltham, MA, USA) at pH 3.6. in a ratio of 75:25 was used. The complex was first dissolved in the mobile phase and then subjected to ultrasonic processing at RT for 40 min. Subsequently, the solution was filtered through a 0.45 µm membrane filter. The HPLC analysis was performed at a flow rate of 0.8 mL/min, with an injection volume of 5 µL, and the total run time was 15 min. The quantification of CBD was carried out by integrating the peak area using a calibration curve of CBD standard (95% purity, THC Pharm GmbH, Frankfurt am Main, Germany). All mobile phase components used were of HPLC quality. The study was performed in 3 replicates.

#### 4.5.2. Chemical Functional Groups and Intermolecular Bonding

The chemical functional groups and intermolecular bonding of CBD, HP-β-CD, and CBD/HP-β-CD complex were assessed using the FTIR technique with the frequency range of 4000−400 cm^−1^. The changes in the FT-IR characteristic peaks of CBD and HP-β-CD regarding the CBD/HP-β-CD interaction were investigated [34].

#### 4.5.3. Surface Morphology

Surface morphology of CBD, HP-β-CD, and CBD/HP-β-CD complex was observed using Field Emission Scanning Electron Microscopy (FE-SEM, Thermo Fisher, Apreo S model, Waltham, MA, USA). Samples were placed onto metal stubs and dried overnight prior to gold deposition to enhance the electrical conductivity. The different morphology of each sample was concerned with observing the formation of CBD/HP-β-CD complexes. The FE-SEM provided topographical and elemental information at magnifications of 10× to 300,000×, with virtually unlimited depth of field.

#### 4.5.4. Crystallinity

The crystallinity of CBD, HP-β-CD, and CBD/HP-β-CD complex was measured using X-ray diffraction (XRD) with the D2 PHASER edition and the LYNXEYE XE-T detector system (Bruker AXS GmbH, Karlsruhe, Germany). The samples were mounted on a vitreous sample holder before the test. Diffraction patterns were recorded with a diffraction angle (2θ) in the range of 10° to 40° (Cu Kα, λ = 1.54 Å).

### 4.6. Fabrication of the Fibroin Film Containing the CBD/HP-β-CD Complex (The Developed Film)

A simple casting technique was employed to produce the film. Following the optimization of the preparation process based on our preliminary study, 600 mg of the isolated fibroin was dissolved in a lactic acid solution (Sigma-Aldrich Chemical GmbH, Steinheim, Germany), and the CBD/HP-β-CD complex was then uniformly dispersed in the fibroin solution (pH 4.0 ± 0.2). The resulting mixture’s final volume was 15 mL. For casting into the film, a 15 mL mixture was poured into a 6 × 6 cm^2^ silicone mold and dried at 47 ± 2 °C for 4 h. Subsequently, the dried film, referred to as the developed film, was gently removed from the mold and stored in a dry and light-protected package until used. The production process of the complex-free fibroin film (referred to as the fibroin film) was similar to the method described above, with the exception that the CBD/HP-β-CD complex was not added.

### 4.7. Characterization of the Developed Film

#### 4.7.1. Determination of the CBD Content in the Film

The quantity of CBD in the film was determined using the HPLC method, as described earlier. The CBD was extracted from the film using the mobile phase and subjected to ultrasonic processing at RT for a minimum of 40 min. The resultant extract solution was then filtered through a 0.45 µm membrane filter before being injected into the HPLC system for analysis. The study was performed in 3 replicates.

#### 4.7.2. Surface Morphology

The surface and cross-sectional morphologies of the fibroin and the developed films were observed using an FE-SEM. The film samples were cut into 1 × 1 cm^2^ and then placed onto an aluminum stub using double-face adhesive tape and dried overnight prior to gold deposition, according to the method described in the previous section. Different areas of the surfaces of films were investigated.

#### 4.7.3. Mechanical Properties

Tensile strength and percentage of elongation at the break of the developed film were determined using a texture analysis (TA.XT Plus, Stable Micro Systems, Ltd., Godalming, UK) in comparison with the fibroin film. The investigation was conducted under specific conditions, including a temperature of 25 °C, humidity of 50%, a load cell of 20 N, a separation speed of 0.5 mm/min, and a sample size of 10 mm × 30 mm × 50 µm. The study was performed in 3 replicates. The calculations for tensile strength and percentage of elongation at break were performed using the following formulas [27]:$$Tensile\;strength\;(MPa)=\frac{Maximum\;force\;(N)\times 100}{Cross-section\;area\;of\;sample\;({mm}^{2})}$$
$$Elongation\;at\;break\;(\%)=\frac{Increase\;in\;length\;(mm)\times 100}{Initial\;length\;(mm)}$$

#### 4.7.4. Chemical Functional Groups and Intermolecular Bonding

Similar to other sections of the FT-IR test, both fibroin film and developed film were measured for their chemical functional groups as well as the possible intermolecular bonding between fibroin and CBD/HP-β-CD complex. The spectra were scanned over a range of wavenumbers from 4000 to 400 cm^−1^.

#### 4.7.5. Crystallinity

The crystallinity of the isolated fibroin, the complex, and the developed film was measured using X-ray diffraction (XRD). The samples were mounted on a vitreous sample holder, and the scan covered a 2θ range from 10° to 40° with a step size. 

### 4.8. Determination of Wound Healing Potential of the Developed Film through Cell-Based Assays

#### 4.8.1. Preparation of the Film-Free Supernatants

The developed and fibroin films were subjected to sterilization using ozone gas [19]. The sterilized films were cut into pieces measuring 1 × 1 cm^2^. They were immersed in 1 mL of serum/antibiotic-free DMEM and incubated at 37 °C for 24 h. The film-free supernatants were further filtered through a 0.22 µm membrane. 

#### 4.8.2. Culture of Normal Human Dermal Fibroblast (NHDF)

NHDF cell lines (Promocell, Eppelheim, Germany) were cultured with passage ranges of 8–15 in Dulbecco’s Modified Eagle Medium (DMEM, Sigma-Aldrich Co.) supplemented with 10% fetal bovine serum (FBS, Gibco, Carlsbad, CA, USA) and 1% penicillin/streptomycin solution (Gibco, Invitrogen, Waltham, MA, USA). Cells were incubated at 37 °C in a humidified atmosphere with 5% CO_2_ and maintained in the culture medium for approximately one week, with regular medium changes every 3 days. The cells exhibiting 80% confluence were used in the experiment.

#### 4.8.3. Cytocompatibility Test

Cell viability of NHDF was measured using 2,3-bis(2-methoxy-4-nitro-5-sulphophenyl)-5-[(phenylamino)carbonyl]-2H-tetrazolium hydroxide (XTT, Roche Diagnostics GmbH, Mannheim, Germany). Briefly, NHDF cells (1 × 10^4^ cells/well) were seeded in a 96-well plate and incubated in DMEM with 10% FBS and 1% antibiotics in a humidified atmosphere with 5% CO_2_ at 37 °C for 24 h. After washing the cells with phosphate-buffered saline (PBS, pH 7.4), they were cultured in the serum/antibiotic-free medium (control group) or treated with the control film (the fibroin film without the CBD/HP-β-CD complex) or developed film-free supernatant for an additional 24 h. Following the treatment, the incubated medium was replaced with a mixture of 200 μL of fresh serum-free medium and 50 μL of XTT reagent. The seeded cells were then incubated at 37 °C for 4 h, and cell viability was determined by measuring the absorbance at 490 nm using a microplate reader. The absorbance of the control group was considered to represent 100% viability. The experiment was performed in 3 replicates. Cell morphology was observed using an inverted microscope.

#### 4.8.4. Cell Cycle Analysis

The cell cycle analysis of cells treated with the film-free supernatant was compared with that of the control group. Briefly, NHDF cells (1 × 10^5^ cells/well) were seeded in a 24-well plate in DMEM with 10% FBS and 1% antibiotics and incubated at 37 °C in 5% CO_2_ incubator for 24 h. After washing the cells with PBS, they were treated with either the control film or developed film-free supernatant for an additional 24 h (control cells were cultured in serum/antibiotic-free medium). Cells were then collected using 0.25% trypsin/0.01 M EDTA (Sigma-Aldrich Co.). The cell suspensions (2 × 10^5^ cells/mL) were centrifuged to obtain cell pellets, which were subsequently washed with PBS and fixed with 70% ethanol. After another washing step, the cells were stained with 150 µL of the Muse^®^ Cell Cycle Assay Kit (Merck KGaA, Darmstadt, Germany) in the dark at RT for 30 min, following the manufacturer’s guidelines. Flow cytometry (Guava^®^ easyCyte^TM^, Merck Millipore, Massachusetts, MA, USA) was utilized to determine the proportion of cells in each cell cycle phase, including G0/G1, S, and G2/M. The experiment was performed in 3 replicates. 

#### 4.8.5. Immunofluorescence of Vascular Epidermal Growth Factor (VEGF)

The anti-VEGFA antibody (ab39250, Abcam, Waltham, MA, USA) was utilized for qualitative evaluation of VEGF expression in NHDF cells. Briefly, NHDF cells (2 × 10^4^ cells/well) were cultured in DMEM with 10% FBS and 1% antibiotics in an 8-well chamber slide at 37 °C for 24 h. After PBS washing, the cells were treated with the control film-free supernatant, the developed film-free supernatant, or the serum/antibiotic-free medium (control) and incubated continuously for 24 h. Following the 24 h incubation, the cells were fixed with 4% paraformaldehyde in PBS at RT for 10 min, permeabilized with 0.1% Triton-X 100 in PBS at RT for an additional 10 min, and then washed three times with PBS (5 min each time). Subsequently, the cells were treated with a blocking solution containing 1% BSA and 22.52 mg/mL glycine in PBS with Tween 20 (PBST) in a humidified environment at 4 °C for 30 min to prevent non-specific antibody binding. The cells were then subjected to overnight incubation with an anti-VEGFA antibody (diluted in 1% BSA in PBST). After five washes with PBS (3 min each time), the cells were incubated in the dark at RT for 1 h with Alexa Fluor^®^ 488 conjugated secondary antibody (diluted in 1% BSA in PBST). The secondary antibody solution was replaced with PBS, and cells were washed five times (3 min each time). Following the washes, the cells were incubated with 100 µL of DAPI (4′,6-diamidino-2-phenylindole, DNA stain) for 10 min, followed by five washes with PBS (3 min each time). Subsequently, the cells were covered with an anti-fade mounting solution and examined with laser scanning confocal microscopy (A1 HD25/A1R HD25, Nikon^®^, Tokyo, Japan) to visualize VEGF expression patterns.

#### 4.8.6. Wound Healing Test

NHDF cells were seeded in the culture-insert 2 well at 2 × 10^5^ cells/mL and incubated in DMEM with 10% FBS and 1% antibiotics for 24 h. After the cells reached 100% confluence, an artificial gap (representing the wound) was created by gently moving the insert well out. Following the removal of the culture media, the cells were washed with 200 µL PBS, and then 1 mL of the developed film-free supernatant or the serum/antibiotic-free medium (control) was added to each well. Images of both the treated and control groups were captured immediately after media replacement (T = 0) and then every 12 h for a total duration of 36 h using microscopy at 10× magnification.

### 4.9. Statistical Analysis

All quantitative data are presented as mean ± standard deviation (S.D.). Group comparisons were performed using Student’s unpaired *t*-test, and statistical significance was considered at *p* < 0.05.

## 5. Conclusions

Our findings suggest that the CBD/HP-β-CD complex could be formed using a molar ratio of 1:2 for CBD to HP-β-CD. This complex could be uniformly dispersed within the prepared aqueous solution of isolated fibroin, enabling the casting of a film with mechanical properties suitable for handling. The XRD pattern of the developed fibroin film containing these complexes showed a characteristic crystalline peak, while the FTIR spectrum exhibited characteristics of both isolated fibroin (amide I, II, and III) and the CBD/HP-β-CD complex. The disappearance of the -C-H stretching peak of the aromatic rings of CBD suggests the effective encapsulation of CBD’s hydrophobic aromatic rings by HP-β-CD. Moreover, the developed film exhibited significant biological activities that contribute to enhanced wound healing. These activities included increased cell division in the G2/M phase, enhanced cell migration, and elevated expression of VEGF in normal human dermal fibroblasts.

## Figures and Tables

**Figure 1 pharmaceutics-15-02682-f001:**
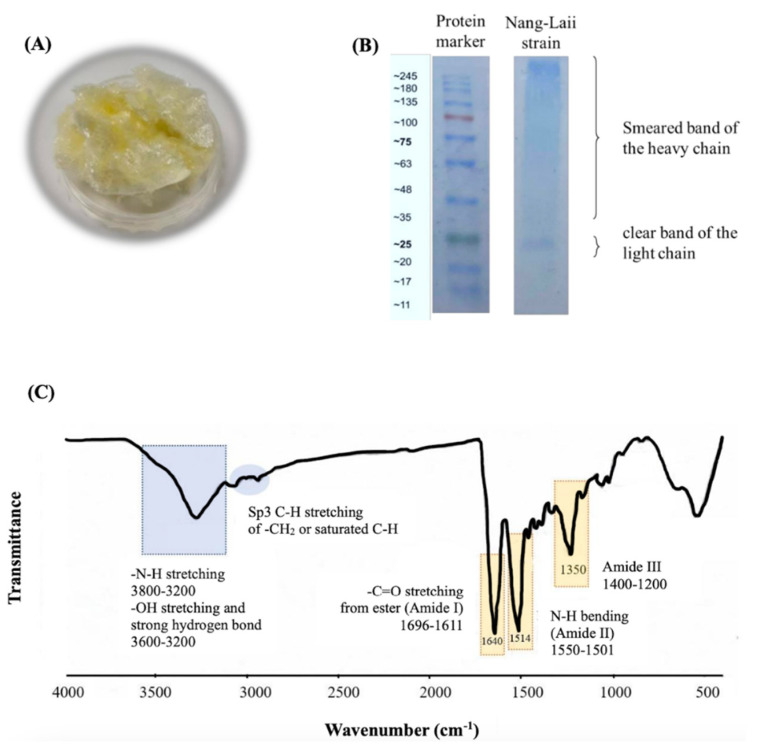
The physical appearance of freeze−dried Nang-Lai strain silkworm cocoon fibroin (**A**), gel electrophoresis for molecular weight determination of isolated fibroin (**B**), and Fourier−transform infrared spectroscopy (FTIR) spectrum of isolated fibroin (**C**).

**Figure 2 pharmaceutics-15-02682-f002:**
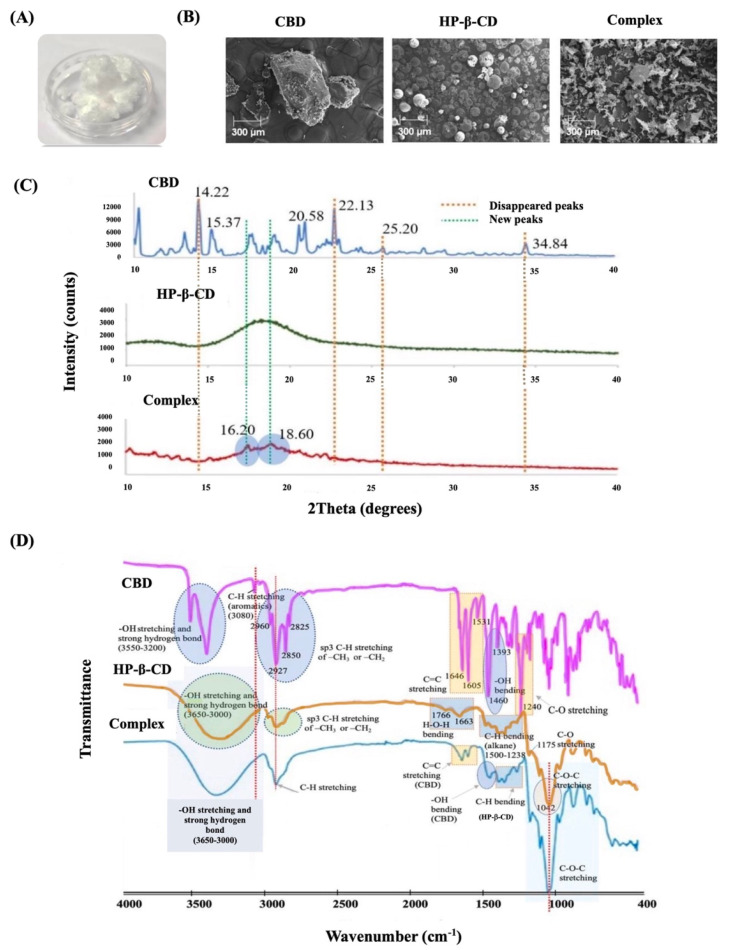
Physical appearance of freeze−dried CBD/HP−β−CD inclusion complex (**A**), field emission scanning electron microscopy (FE−SEM) images (magnification ×100) (**B**), X−ray diffraction (XRD) patterns (**C**), and Fourier−transform infrared spectroscopy (FTIR) spectra (**D**) of CBD, HP−β−CD, and CBD/HP−β−CD inclusion complex.

**Figure 3 pharmaceutics-15-02682-f003:**
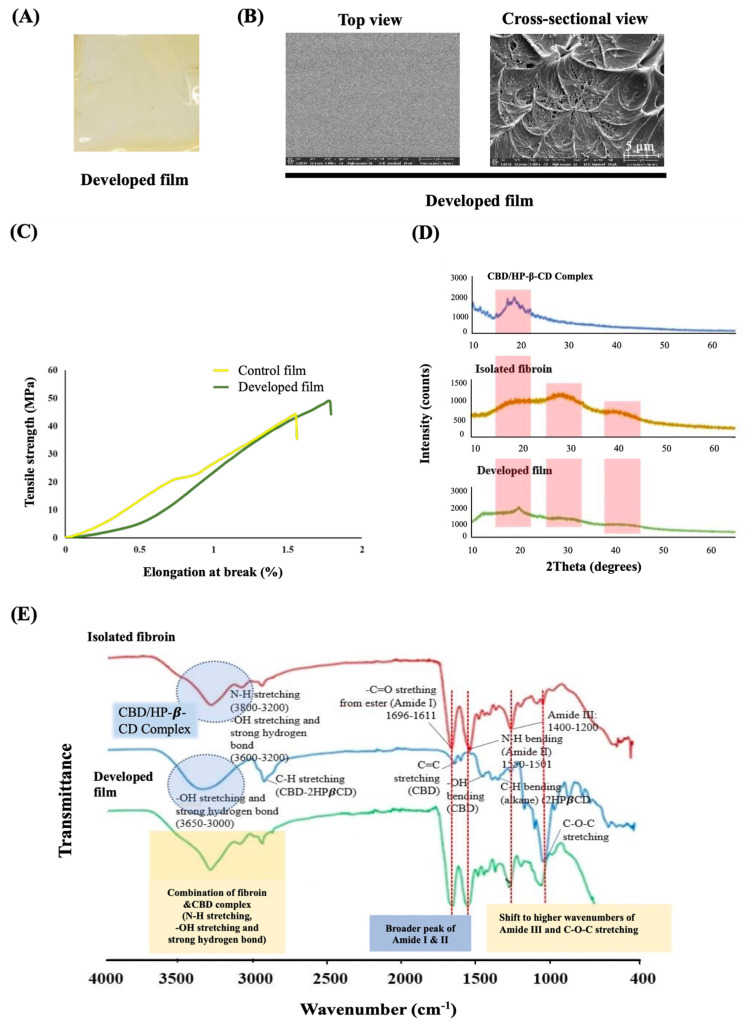
Physical appearance of developed film (fibroin film with CBD/HP−β−CD inclusion complex) (**A**), field emission scanning electron microscopy (FE−SEM) images at 5000× magnification of the film surface (**B**), tensile properties including strength and elongation at break of developed film and control film (**C**), X−ray diffraction (XRD) patterns (**D**), and Fourier-transform infrared spectroscopy (FTIR) spectra of CBD/HP−β−CD complex, isolated fibroin, and the developed film (**E**).

**Figure 4 pharmaceutics-15-02682-f004:**
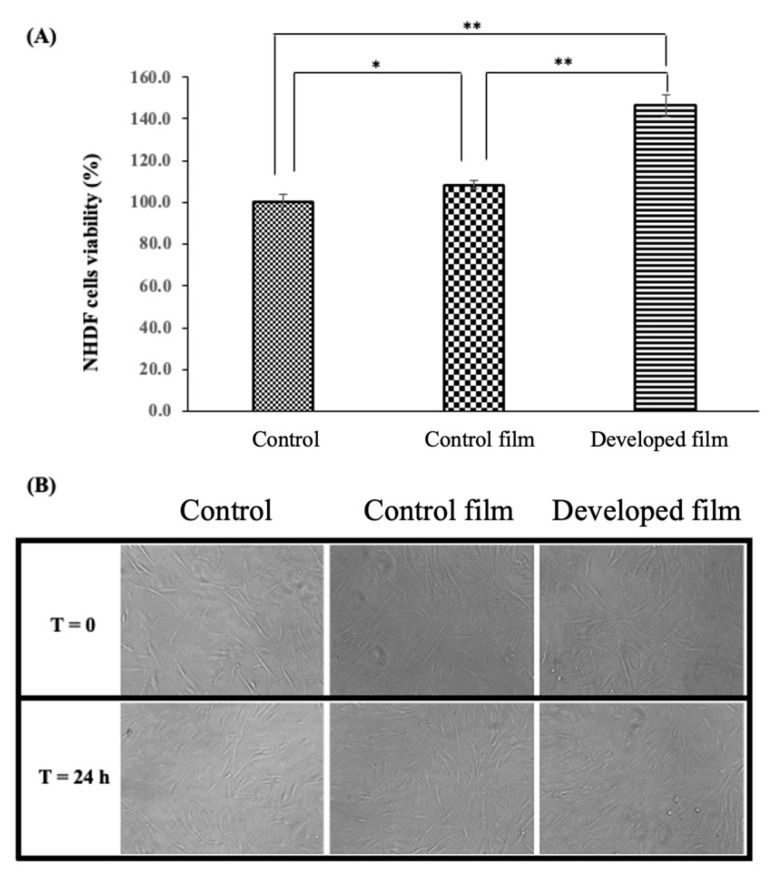
Viability and morphology of normal human dermal fibroblast (NHDF) cells treated with the control film or the developed film-free supernatant for a duration of 24 h. The viability of NHDF cells was determined by XTT assay and, in comparison, to the serum/antibiotic-free DMEM (control group, 100% viability) (**A**) and cell morphology of NHDF cells at a magnification of 20× (**B**). Each bar represents mean ± S.D., n = 3, * *p* < 0.05, ** *p* < 0.01.

**Figure 5 pharmaceutics-15-02682-f005:**
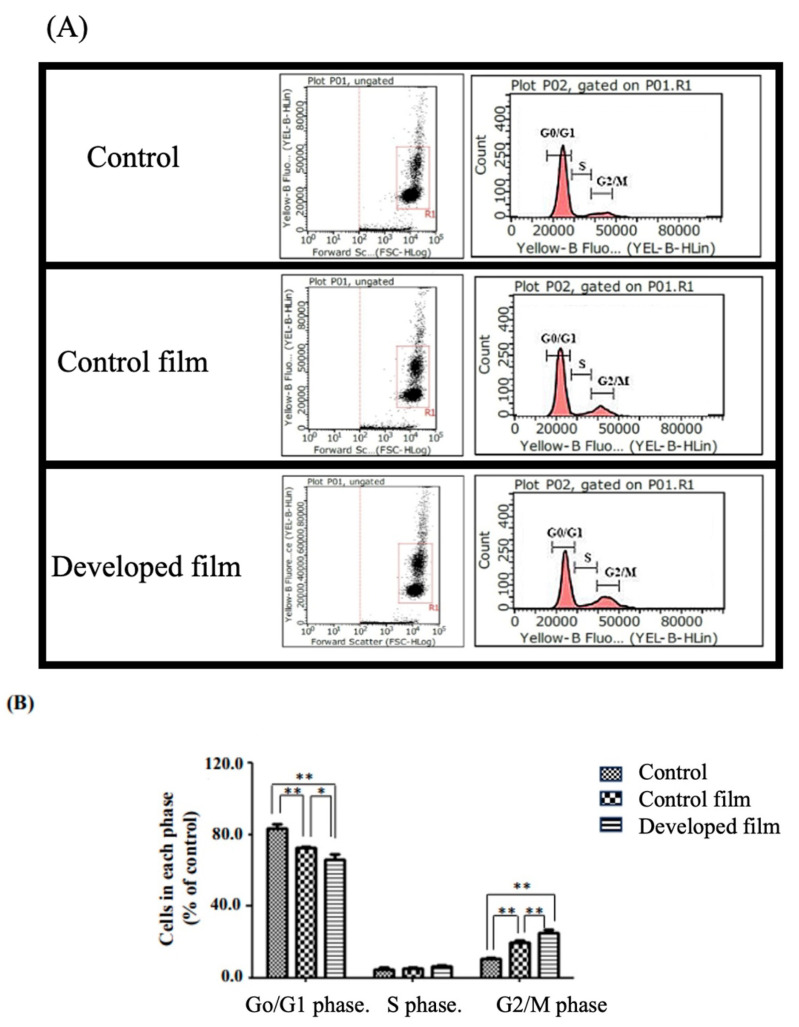
The cell cycle phases of normal human dermal fibroblast (NHDF) cells were treated with the control film and the developed film-free supernatant for a duration of 24 h, in comparison to the control (the serum/antibiotic-free DMEM) group. Dot plots and histogram profiles are used to visualize the cell cycle distribution (**A**) and percentage of the total cells in each phase (**B**). Each bar represents mean ± S.D., n = 3, * *p* < 0.05, ** *p* < 0.01.

**Figure 6 pharmaceutics-15-02682-f006:**
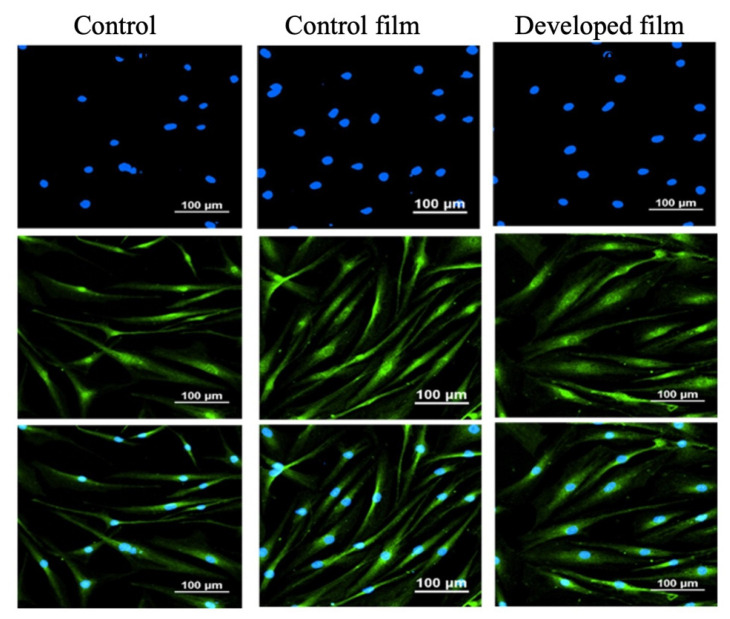
Immunofluorescence images of vascular endothelial growth factor (VEGF) protein (green) and nuclei (blue) in normal human dermal fibroblast (NHDF) cells treated with the control film and the developed film-free supernatant for 24 h, in comparison to the control (the serum/antibiotic-free DMEM) group captured by confocal microscopy at a magnification of 20×.

**Figure 7 pharmaceutics-15-02682-f007:**
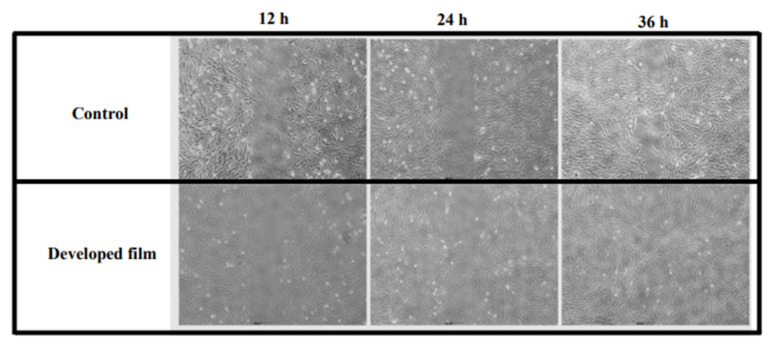
The cell migration of NHDF cells treated with the developed film-free supernatant for 12, 24, and 36 h, in comparison to the control (the serum/antibiotic-free DMEM) group, at a magnification of 10×.

## Data Availability

The raw/processed data required to reproduce these findings cannot be shared at this time as the data also forms part of an ongoing study.

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
