# Peer review of "In Vitro Wound Healing Potential of a Fibroin Film Incorporating a Cannabidiol/2-Hydroxypropyl-β-cyclodextrin Complex"

_pharmaceutics, 2023, doi:10.3390/pharmaceutics15122682_

Round 1

Reviewer 1 Report

Comments and Suggestions for Authors

The article reads well and I only have minor comments:

1. The title does not read well. I would omit enhancement from the title or rephrase it

2. Figure legends are fuzzy. The figure axes should have the same font and font size

3. Figure 5 should be replaced by one that is clearer

4. The Discussion should start by a summary statement of cardinal findings

5. Materials and Methods: add manufacturer information to all materials

6. Introduction and Discussion: add references to support claims from previous reports

7. A native English scientist should proof read the whole manuscript.

Comments on the Quality of English Language

Proofreading is needed

Author Response

  1. The title does not read well. I would omit enhancement from the title or rephrase

Answer: As suggested, the manuscript title has been changed to “In vitro wound healing potential of a fibroin film incorporating a cannabidiol/2-hydroxypropyl-β-cyclodextrin complex”

  1. Figure legends are fuzzy. The figure axes should have the same font and font size

Answer: As suggested, the figures and figured legend were revised and modified.

  1. Figure 5 should be replaced by one that is clearer.

Answer: As suggested, the figures and figure legends have been improved for better clarity and understanding.

  1. The Discussion should start by a summary statement of cardinal findings

Answer: As suggested, the discussion part starts with a summary statement of findings (from line 254 to 261).

  1. Materials and Methods: add manufacturer information to all materials

Answer: As suggested, the manufacturer information has been included in all relevant materials.

  1. Introduction and Discussion: add references to support claims from previous reports

Answer: As requested, references have been added to support claims from previous reports.

  1. A native English scientist should proof read the whole manuscript.

Answer: As requested, our co-author, a native English scientist, has proofread the manuscript.

Reviewer 2 Report

Comments and Suggestions for Authors

1. These elements (2-hydroxypropyl, CBD, and β-cyclodextrin) added by the authors in the material design process and their respective advantages should be explained. In this way, the reader will have a clearer idea of the features of this article.

2. The Conclusion section should be strengthened: the important results and main conclusions drawn in this paper should be highlighted and presented in more precise language.

3. Background descriptions for wound healing can be strengthened by citing 10.1002/adma.202306632; 10.1016/j.ccr.2023.215426 and what are the advantages of the current work compared to published articles?

4. Some of the images in the article need to be re-presented. For example, Figure 5, the font in the image is too small and hard to read.

5. Where are these dressing films going to be used in real life? Advantages of the designed CBD/HP-β-CD can be improved by comparing clinical products. It could be better if a brief comment (challenges and future prospects) is added at the manuscript.

Comments on the Quality of English Language

N/A

Author Response

Reviewer 2

  1. These elements (2-hydroxypropyl, CBD, and β-cyclodextrin) added by the authors in the material design process and their respective advantages should be explained. In this way, the reader will have a clearer idea of the features of this article.

Answer: As suggested, the introduction section now elaborates on the respective advantages of CBD and 2-hydroxypropyl-β-cyclodextrin (HP-β-CD) from line 93 to 100.

  1. The Conclusion section should be strengthened: the important results and main conclusions drawn in this paper should be highlighted and presented in more precise language.

Answer: As suggested, the conclusion section has been revised by adding the important results and highlight of the findings (from line 576 to 587).

  1. Background descriptions for wound healing can be strengthened by citing 10.1002/adma.202306632; 10.1016/j.ccr.2023.215426 and what are the advantages of the current work compared to published articles.

Answer: As suggested, we have cited the published article (10.1002/adma.202306632) that pertains to the hydrogel incorporating a compound synthesized from melanin/gold/platinum, which successfully promoted wound healing in diabetic rats by facilitating hyperthermia and disrupting the ROS-inflammation pathway (from line 72 to 77).

  1. Some of the images in the article need to be re-presented. For example, Figure 5, the font in the image is too small and hard to read.

Answer: As suggested, the figures and figure legends have been improved for better clarity and understanding.

  1. Where are these dressing films going to be used in real life? Advantages of the designed CBD/HP-β-CD can be improved by comparing clinical products. It would be better if a brief comment (challenges and future prospects) is added at the manuscript.

Answer: As suggested, a concise comment on challenges and prospects has been added into the discussion section of the manuscript (from line 365 to 370).

Reviewer 3 Report

Comments and Suggestions for Authors

1.     the title should include the full name of cannabidiol instead of CBD

2.     The resolution of all figures needs to be improved. Most of the figures don't even allow you to read their content. Figure 5 shows neither the scale nor the descriptions of the axes, much less the results.

3.     A scratch test should be used to assess wound healing potential. However, in Figure 4 there is no visible scratch.

4.     Chapter 3 is missing a significant portion of literature references. Lines 255-280 refer to only two references.

5.     Why is there no comment in the results section regarding the preparation of the system itself? Was the immediately selected lineup the final one? Have any preliminary tests been performed?

6.     What about assessing anti-inflammatory effects?

Author Response

Reviewer 3

  1. the title should include the full name of cannabidiol instead of CBD­

Answer: As suggested, the manuscript title has been changed to “In vitro wound healing potential of a fibroin film incorporating a cannabidiol/2-hydroxypropyl-β-cyclodextrin complex”

  1. The resolution of all figures needs to be improved. Most of the figures don't even allow you to read their content. Figure 5 shows neither the scale nor the descriptions of the axes, much less the results.

Answer: As suggested, the figures and figure legends have been improved for better clarity and understanding.

  1. A scratch test should be used to assess wound healing potential. However, in Figure 4 there is no visible scratch.

Answer: The results of the scratch test have been shown in Figure 7.

  1. Chapter 3 is missing a significant portion of literature references. Lines 255-280 refer to only two references.

Answer: As suggested, additional relevant references have been included throughout the manuscript.

  1. Why is there no comment in the results section regarding the preparation of the system itself? Was the immediately selected lineup the final one? Have any preliminary tests been performed?

Answer: The preparation of the designed system followed the optimization of the preparation process based on our preliminary study, as mentioned in lines 512 to 513 (“Fabrication of the fibroin film containing the CBD/HP-β-CD complex”).

  1. What about assessing anti-inflammatory effects?

Answer: In this study, we did not investigate the anti-inflammatory activity of the developed film using inflammatory cells. Instead, our primary focus was on investigating the potential use of the developed film in dermal fibroblast cultures, as fibroblasts play a central role in the wound healing process [2], as mentioned in line 322 to 323. Nevertheless, we acknowledge and agree with your recommendation. To ensure the effective wound healing properties of the developed film, further studies involving inflammatory-inducing model in vitro and/or in vivo should be conducted in the future.

Round 2

Reviewer 3 Report

Comments and Suggestions for Authors

Accept in present form

Author Response

thank you